# *Plasmodium* SAS4: basal body component of male cell which is dispensable for parasite transmission

Mohammad Zeeshan[1,2,*], Declan Brady[1,*], Robert Markus[1], Sue Vaughan[3], David Ferguson[3], Anthony A Holder[4], Rita Tewari[1]

The centriole/basal body (CBB) is an evolutionarily conserved organelle acting as a microtubule organising centre (MTOC) to nucleate cilia, flagella, and the centrosome. SAS4/CPAP is a conserved component associated with BB biogenesis in many model flagellated cells. *Plasmodium*, a divergent unicellular eukaryote and causative agent of malaria, displays an atypical, closed mitosis with an MTOC (or centriolar plaque), reminiscent of an acentriolar MTOC, embedded in the nuclear membrane. Mitosis during male gamete formation is accompanied by flagella formation. There are two MTOCs in male gametocytes: the acentriolar nuclear envelope MTOC for the mitotic spindle and an outer centriolar MTOC (the basal body) that organises flagella assembly in the cytoplasm. We show the coordinated location, association and assembly of SAS4 with the BB component, kinesin-8B, but no association with the kinetochore protein, NDC80, indicating that SAS4 is part of the BB and outer centriolar MTOC in the cytoplasm. Deletion of the *SAS4* gene produced no phenotype, indicating that it is not essential for either male gamete formation or parasite transmission.

## Introduction

Centriole/basal bodies (CBBs) are associated with the microtubule organising centre (MTOC) that nucleates cilia, flagella, and centrosomes and are conserved ancestral organelles in eukaryotes (Carvalho-Santos et al, 2011; Nabais et al, 2020). Centrioles and basal bodies (BBs) share structural features and BBs are mainly associated with flagella or cilia organisation, and extend to produce an axoneme (Marshall, 2008). The canonical view of CBB biogenesis includes centriole duplication and segregation to a daughter cell during mitosis. However, in some organisms, BB biology is more diverse, for example, where the centrioles or BBs form de novo or exhibit non-canonical biogenesis, as observed in Naegleria and in some parthenogenic insect eggs (Fritz-Laylin & Fulton, 2016; Nabais et al, 2017, 2020). SAS4/SAS6 are ancestral core proteins involved in BB biogenesis, as predicted by phylogenetic analysis (Carvalho-Santos et al, 2010; Hodges et al, 2010).

*Plasmodium spp.*, the causative agents of malaria, are apicomplexan parasites transmitted by mosquito vectors. Asexual replication in *Plasmodium* occurs by atypical closed endomitosis, with remarkable plasticity in unconventional aspects of cell division during its complex life cycle. In a crucial stage for parasite transmission, sexually committed cells—the male and female gametocytes—are formed in the mammalian host and activated following ingestion by the female mosquito vector during its blood meal. Activation in the midgut results in gametogenesis and formation of extracellular female and flagellate male gametes over a period of 15 min (Sinden, 1991; Sinden et al, 2010). During male gametogenesis within 8 min, there are three rounds of genome replication from haploid (1N) to octaploid (8N) without nuclear division and this is followed by karyokinesis and cytokinesis resulting in eight haploid flagellated gametes in a process known as exflagellation (Sinden et al, 2010). Flagella assembly is very rapid and atypical, occurring within 15 min and without intra-flagellar transport (Sinden, 1991; Sinden et al, 2010). There are no flagella at other stages of the life cycle and hence no clear centriole or BB is observed at other proliferative life cycle stages. Clear centrioles with 9 + 1 or 9 + 2 microtubules can only be seen in flagella biogenesis during male gamete formation in *Plasmodium* (Sinden et al, 2010; Straschil et al, 2010; Zeeshan et al, 2019a). So-called centriolar plaques located within the nuclear envelope were described during nuclear division in asexual stages of proliferation and appear to serve as an MTOC in nuclear spindle formation (Arnot et al, 2011; Gerald et al, 2011). Centrin has been mapped to these plaques and used as a marker for the MTOC to follow the asynchronous replication dynamics during asexual replication. However, no centriole is present, and these plaques resemble an acentriolar MTOC organising hemispindle (Mahajan et al, 2008; Roques et al, 2019; Bertiaux et al, 2021; Simon et al, 2021). In male gametocytes, there are two MTOCs: an inner acentriolar MTOC located with the nuclear

[1]School of Life Sciences, University of Nottingham, Nottingham, UK   [2]Faculty of Infectious and Tropical Diseases, London School of Hygiene and Tropical Medicine, London, UK   [3]Department of Biological and Medical Sciences, Faculty of Health and Life Science, Oxford Brookes University, Oxford, UK   [4]Malaria Parasitology Laboratory, The Francis Crick Institute, London, UK

Correspondence: rita.tewari@nottingham.ac.uk; mohammad.zeeshan@lshtm.ac.uk
*Mohammad Zeeshan and Declan Brady contributed equally to this work.

envelope, and outer centriolar BB located within the cytoplasm and required for flagellum assembly. The marker, NDC80 shows that kinetochores are clustered in the nucleus in asexual stages with a rod-like structure at spindle formation during the three successive rounds of genome replication during in gametogenesis, which can be differentiated from the outer MTOC (Zeeshan et al, 2020b).

BB structure and the real time dynamics of its formation during this accelerated genome replication and chromosome segregation remain unclear in Plasmodium, although earlier ultrastructural studies by EM suggested atypical BB structure and flagella (Sinden et al, 1976). Ultrastructure studies also confirmed that the nuclear envelope remains intact during the closed mitosis, so events in the nucleus and cytoplasm must be coordinated for flagellated gamete formation to occur: the two MTOCs need to be organised and co-ordinated between these compartments so that each male gamete receives a single flagellum. In recent studies we have shown that some molecular motors like kinesin-8X and kinesin-5 are associated with the spindle in the nuclear compartment (Zeeshan et al, 2019b, 2020a) and others like kinesin-8B and kinesin-X4 are involved in axoneme biogenesis (Zeeshan et al, 2019a). Therefore, the MTOCs during male gametogenesis may be described as composed of an outer centriolar MTOC, which organises the BB and axoneme linked with an inner acentriolar MTOC similar to the spindle pole body of yeast from where the mitotic spindle and chromosome segregation is organised (Zeeshan et al, 2019a, 2019b). An earlier study with the BB marker SAS6 showed it was present outside the nucleus in the cytoplasmic BB compartment during male gametogenesis and its deletion ablated male gametogenesis, blocking parasite transmission (Marques et al, 2015). Recently, we and others have shown the dynamic profile of kinesin-8B, another BB marker, in axoneme assembly and its deletion leads to disruption of axoneme assembly and loss of flagellum formation (Zeeshan et al, 2019a; Depoix et al, 2020).

Because SAS4 and SAS6 are both ancestral components of BB formation (Carvalho-Santos et al, 2010, 2011; Hodges et al, 2010), here we have investigated the real time temporal profile of SAS4 during male gametogenesis, particularly during the 8 min of rapid genome replication following activation. We also investigated its subcellular association with kinetochore and axoneme biogenesis by genetically crossing SAS4-GFP and NDC80-mCherry or kinesin-8B-mCherry transgenic parasite lines to obtain male gametocytes expressing both fluorescent markers. The results reveal a de novo rapid synthesis of SAS4 and suggest an association with the outer centriolar BB as a doublet during male gametogenesis. To examine the functional role of SAS4, we generated SAS4 gene KO parasites. However, in contrast to the SAS6 KO mutant, the SAS4 KO mutant developed normally, indicating that this protein is not essential for parasite growth in the mosquito or transmission.

# Results

## Live cell imaging of SAS4-GFP reveals de novo BB formation and its rapid dynamics during male gametogenesis

To analyse the expression of SAS4, first we examined the transcript level of SAS4 in asexual erythrocytic and gametocyte stages. To quantify the transcript level of SAS4, we isolated RNA from respective stages and performed qRT-PCR. We found that SAS4 is expressed mainly in gametocyte stages although its transcript is also detected in asexual erythrocytic stages (Fig S1A).

To study the location of SAS4 during the Plasmodium life cycle, we generated a C-terminal tagged SAS4-GFP transgenic line using rodent malaria model Plasmodium berghei, by inserting an in-frame gfp coding sequence at the 3′ end of the endogenous sas4 locus using single homologous recombination (Fig S1B). PCR analysis of genomic DNA using locus-specific diagnostic primers indicated correct integration of the GFP tagging construct (Fig S1C). We confirmed the expression of SAS4-GFP by Western blotting using anti-GFP antibody (Fig S1D). This transgenic line was used to examine the spatiotemporal profile of SAS4-GFP protein expression and location by real time live cell fluorescence imaging and fixed immunofluorescence assay. SAS4 was not detectable in asexual blood stages and female gametocytes (Fig S2A and B) but was located in the cytoplasm of male gametocytes (Figs 1 and S2B). Therefore, we investigated its expression and location throughout male gametogenesis. Male gametogenesis is a rapid process of three rounds of genome replication, de novo BB formation and axoneme assembly, followed by emergence of eight flagellated male gametes in a process known as exflagellation, which completes within 12–15 min (Sinden et al, 2010).

Live cell images showed the de novo appearance of multiple discrete SAS4-GFP foci in the cytoplasm of activated gametocytes, with the number of foci depending on the length of time after activation (Fig 1A). To further resolve the SAS4-GFP foci, 3D-structured illumination microscopy (SIM) was performed on fixed gametocytes expressing SAS4-GFP (Fig 1B). Live cell and SIM images showed that within 1 minute post-activation (mpa) of the gametocyte, four closely associated foci forming an SAS4-GFP tetrad were observed in the cytoplasm at one side of the nucleus (Fig 1A and B). The SAS4-GFP tetrad split later into two halves that moved apart to opposite sides of the nucleus within 3 mpa, still retaining the cytoplasmic location (Fig 1A–C and Video 1). The two SAS4-GFP tetrads each split again into two doublets of SAS4-GFP and separated from each other within 4–5 mpa (Fig 1A, B, and D and Video 2). A final round of splitting and separation occurred to produce eight discrete SAS4-GFP foci (Fig 1A) leading to formation of flagellated male gametes showing a discrete focal point of SAS4-GFP at the end of the flagellated gamete (Fig 1A). A schematic diagram of this process is provided in the upper panel of Fig 1A.

## SAS4 associates with kinesin-8B, a molecular motor that regulates BB formation and axoneme assembly

Recently we showed that kinesin-8B associates with the BBs and axonemes during Plasmodium male gametogenesis (Zeeshan et al, 2019a); live cell imaging showed the association of kinesin-8B with the tetrad of BBs that serve as a template for axoneme assembly (Zeeshan et al, 2019a). To establish whether SAS4 is part of the BB and associated with tetrad of BB formation and axoneme assembly, we examined its location compared with that of kinesin-8B. A parasite line expressing both SAS4-GFP and kinesin-8B-mCherry was produced and used for live cell imaging by fluorescence microscopy to establish the spatiotemporal relationship of these two

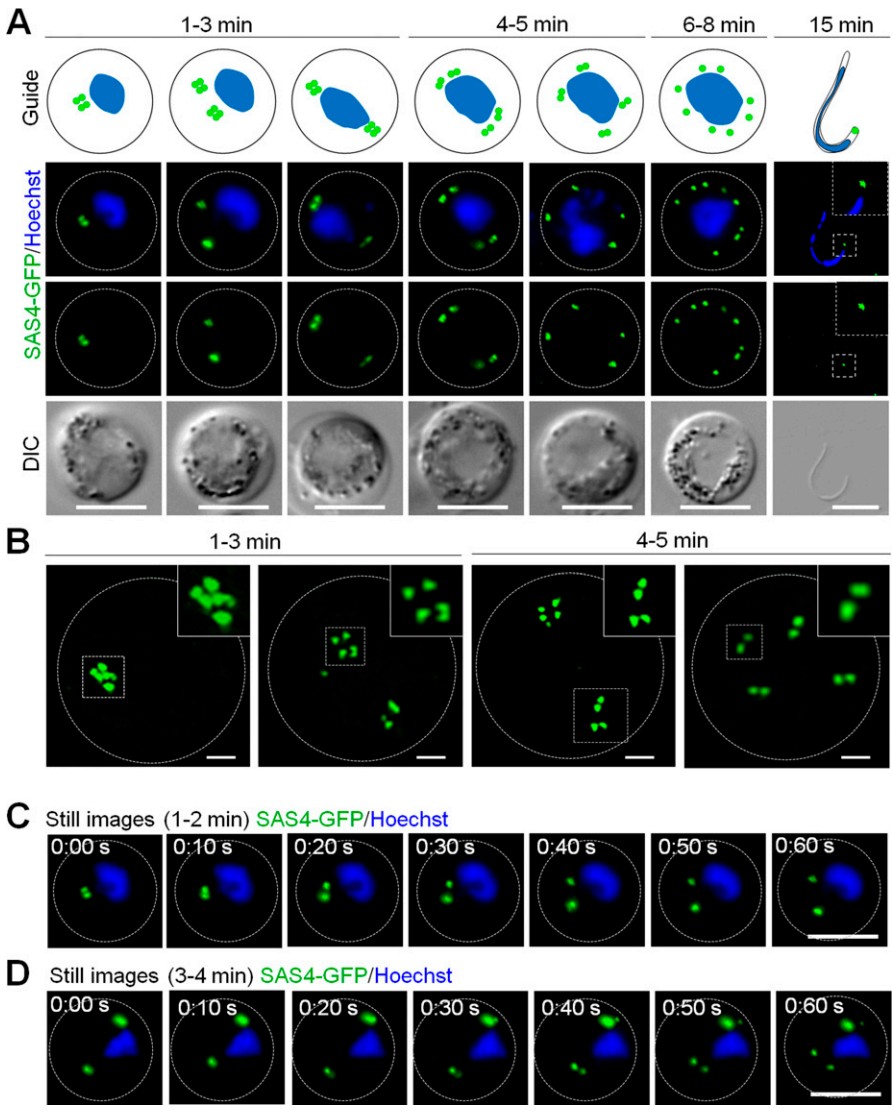

**Figure 1.  Localization of SAS4-GFP during male gametogenesis.**
**(A)** Live cell imaging of SAS4-GFP expression and location during endoreduplicative mitotic division in male gametogenesis. Scale bar = 5 μm. Schematic guide showing SAS4-GFP location in relation to nucleus (DNA) during male gametogenesis. **(B)** Super-resolution 3D imaging for SAS4-GFP localization in gametocytes fixed at 1–3- and 4–5 min post-activation. Scale bar = 1 μm. **(C, D)** Time-lapse screenshots for SAS4-GFP localization in male gametocytes at 1–2 min (C) and 3–4 min (D) post-activation during male gametogenesis. Scale bar = 5 μm (D).

proteins. Within 1 mpa, the SAS4-GFP tetrad was observed at the centre of four kinesin-8B-mCherry foci in the cytoplasm at one side of the nucleus (Fig 2A). Within 3 mpa, we observed the duplication and separation of tetrads of SAS4 and kinesin-8B (Fig 2A). To further resolve this dissociation, we performed 3D-SIM on fixed gametocytes expressing these two proteins. 3D-SIM images clearly show the SAS4 tetrads at the centre of kinesin-8B tetrads (Fig 2A, right hand panel). After arrival at either side of the nucleus, the emergence of axonemes was observed, as revealed by kinesin-8B that later is only associated with axonemes (Fig 2A). The SAS4-GFP tetrad later split into doublets, which remain associated with growing axonemes labelled with kinesin-8B-mCherry during their further split and separation (Fig 2A). At the end of the process, we observed eight SAS4-GFP foci associated with fully assembled axonemes (Fig 2A). These data show the association of SAS4 with kinesin-8B through nonoverlapping distinct, though connected, locations, during a very early stage of BB formation, and remains associated with it throughout axoneme assembly during the rest of male

gametogenesis. A schematic diagram of this process is provided on the top of left panel of Fig 2A.

### The spatiotemporal locations of SAS4 and the kinetochore protein NDC80 reveal BB formation and mitotic spindle dynamics are coupled processes

During mitosis in male gametogenesis, genome replication and chromosome segregation are rapid processes. To determine the relationship between mitosis in the nucleus and BB formation in the cytoplasm, a parasite line expressing both SAS4-GFP, and kinetochore protein NDC80-mCherry was used to image these markers in the same cell. Within 1 mpa, the SAS4-GFP tetrad and the NDC80-mCherry focal point were adjacent but not overlapping close to the nuclear DNA (Figs 2B and S2C), with SAS4 in a cytoplasmic location and NDC80 closer to the DNA. Later in gametogenesis the SAS4 tetrad split into two parts with the NDC80 signal extending to form a bridge between them, which is presumably the

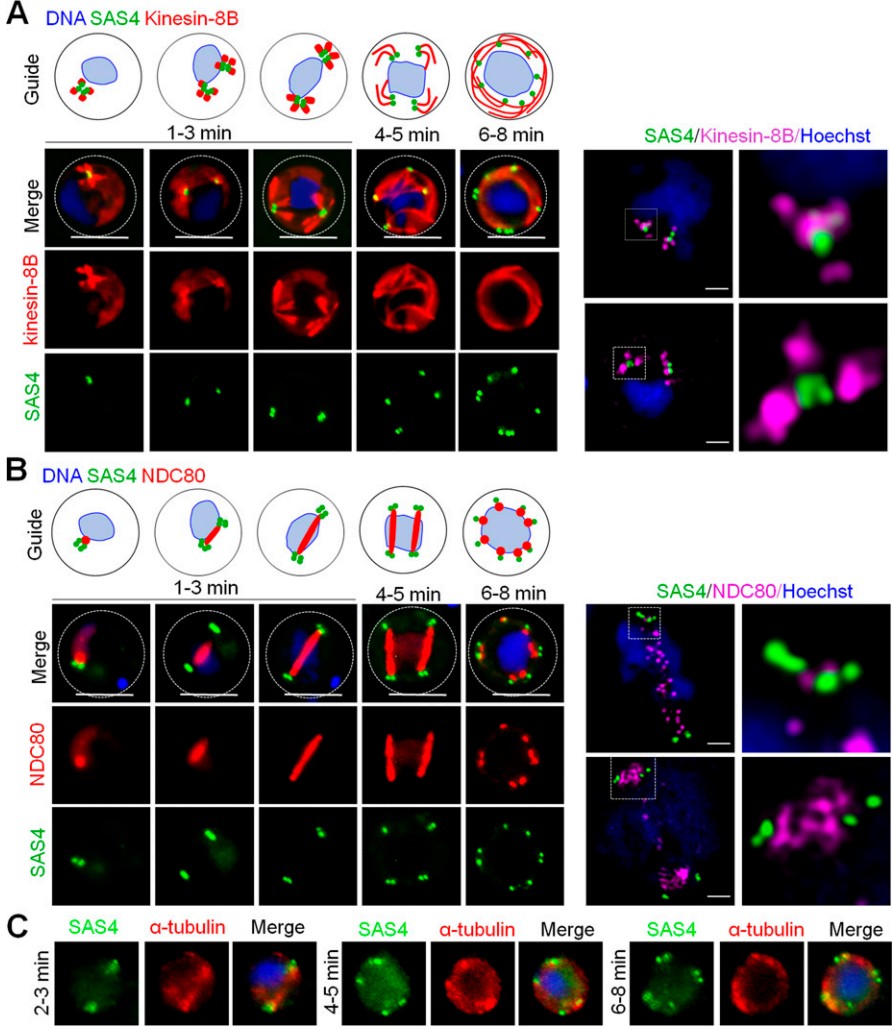

**Figure 2. The location of SAS4 in relation to that of the basal body (BB) and axoneme (kinesin-8B) and kinetochore (NDC80) markers.**
**(A)** The location of SAS4-GFP (green) in relation to the BB and axoneme marker, kinesin-8B–mCherry (red) during male gamete formation. SAS4 shows a cytoplasmic location like kinesin-8B and remains associated with it during BB biogenesis and axoneme formation throughout male gamete formation. Scale bar = 5 μm. Right hand panel shows super-resolution 3D imaging for SAS4-GFP and kinesin-8B–mCherry localization in gametocytes fixed at 1–2 min post-activation. Scale bar = 1 μm. A schematic is shown on top that corresponds to left panel. **(B)** The location of SAS4-GFP (green) in relation to the kinetochore marker, NDC80-mCherry (red) during male gamete formation. The cytoplasmic location of SAS4 contrasts with the nuclear location of NDC80 during chromosome replication and segregation, indicating that SAS4 is not associated with the mitotic spindle. Scale bar = 5 μm. Right hand panel shows super-resolution 3D imaging for SAS4-GFP and NDC80-mCherry localization in gametocytes fixed at 2–3 min post-activation. Scale bar = 1. A schematic is shown on top that corresponds to left panel. **(C)** Fixed-cell immunofluorescence assay showing location of SAS4 in relation to microtubules. Scale bar = 5 μm.

mitotic spindle decorated with kinetochores (Figs 2B and S2C). As the two SAS4 tetrads moved apart the NDC80-positive bridge extended across one side of the nucleus and then separated into two halves (Fig 2B). To resolve further the location of SAS4 tetrads and the NDC80 bridge, we used 3D-SIM on fixed gametocytes expressing these two labelled proteins. The 3D-SIM images clearly showed the two SAS4 tetrads at both ends of the NDC80-positive bridge that then divides into two halves (Fig 2B, right hand panel). The two halves of the NDC80-positive bridge further extend to form two bridges, along with concurrent separation of the SAS4 tetrads into doublets (Figs 2B and S2C). This process of NDC80-positive bridge formation and separation continues for a third cycle, resulting in eight NDC80 and SAS4 foci (Fig 2B). During the whole process of NDC80-labelled bridge formation and separation, SAS4 was located adjacent to but never overlapped with NDC80 (Fig 2B). A schematic diagram for this process is provided on the top of left panel of Fig 2B.

We also performed fixed immunofluorescence assays using anti-GFP and anti-tubulin antibodies to study the location of SAS4 in relation to microtubules in gametocytes. We observed clear punctate locations of SAS4 that do not colocalise with microtubules during male gametogenesis (Fig 2C).

## EM analysis suggests that SAS4 is part of an outer centriolar BB MTOC in male gametocytes

From the literature it is unclear whether the acentriolar MTOC located at the electron dense nuclear pole at the nuclear membrane and BB centriolar MTOC located in the cytoplasm are linked and performing the same role or there are two independent MTOCs, one that is organising the spindle dynamics in the nucleus and other organising the axoneme biogenesis. Therefore we examined various transmission electron micrographs of male gametocytes and compared them with some of the images of *Plasmodium yoelii* male gametocytes described earlier by Sinden et al (1976). Our micrographs support the relative location of BB, axoneme, nucleus and kinetochore, as described by Sinden et al (1976). In these micrographs, two adjacent electron dense masses are observed on either side of the nuclear membrane (Figs 3A and S2). The outer electron dense mass has occasionally nine single α-tubules (Fig 3Bi

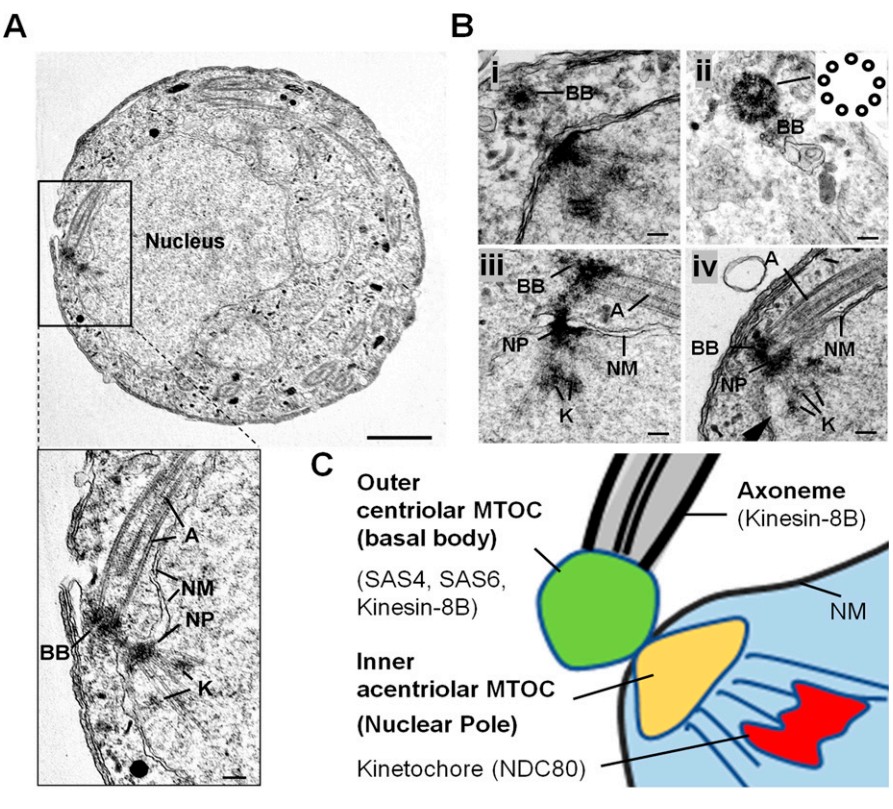

**Figure 3. SAS4 is a part of the outer centriolar MTOC (basal body [BB]).**
**(A)** EM on 4–5 min post-activation gametocyte reveals the relative locations of BB, nuclear pole, and kinetochore. Section through the microgametocyte showing a large central nucleus (N) with BB in the peripheral cytoplasm. Scale bar = 1 $\mu$m. Enlargement of the enclosed area showing the details of BB with attached axoneme (A) in the cytoplasmic compartment separated by a nuclear membrane (NM) from an intranuclear spindle with attached kinetochores (K) radiating from the nuclear poles (NP). Scale bar = 100 nm. **(B)** Enlarged electron micrographs of gametocytes showing the dense BB structures (i and ii). Scale bar = 100 nm. Enlarged electron micrographs of gametocytes (iii and iv) showing the details of BB and axoneme (A) in cytoplasmic compartment separated by a nuclear membrane (NM), an intranuclear spindle with attached kinetochores (K) radiating from the nuclear poles (NP). Scale bar = 100 nm. **(C)** A schematic diagram showing outer centriolar MTOC (BB) and inner acentriolar MTOC (Nuclear Pole) serving as microtubule organising centres for axoneme and intranuclear spindle respectively.

NM, Nuclear Membrane; BB, Basal Body; NP, Nuclear Pole; K, Kinetochore; A, Axoneme

and ii), typical of that report for the BB in other Apicomplexa (Francia et al, 2015). The inner electron dense part is a nuclear pole (Fig 3Biii and iv). Both structures serve as MTOCs: the BB for axoneme microtubules (MTs) and the nuclear pole for spindle MTs to which kinetochores are attached (Fig 3A and B). During mitosis in the asexual blood stages there is no BB but the MTOC for mitotic spindle MTs is present and located within the nuclear envelope. This observation is consistent with the location of two separate and distinct MTOC. The first one is the nuclear pole (NP) in gametocytes that serves as an inner acentriolar MTOC for spindle MTs and the second is where SAS4, SAS6, and kinesin-8B are located in the BB and part of outer centriolar MTOC (Fig 3C). These two independent MTOC have to be coordinated for the successful generation of flagellate male gametes.

### SAS4-GFP shows discrete foci at apical end during zygote to ookinete development and has a nuclear location during meiosis

Because the SAS4-GFP showed a punctate location at the apical end of flagellated male gametes and is absent from female gametes, we were tempted to study its expression and localization in zygotes following fertilization. By live cell imaging we observed that SAS4-GFP has two discrete foci next to the nucleus in the zygote, which separate from each other in the later stages (6–8 h) of ookinete development (Fig S2D). A strong SAS4-GFP fluorescence signal was also observed at the apical end of retorts (8–10 h) during ookinete

development (Fig S2E). In mature ookinetes (24 h), the SAS4-GFP signal disappeared from both nucleus and apical end (Fig S2D). Analysis of a parasite line expressing both SAS4-GFP and NDC80-mCherry showed that SAS4-GFP has a peripheral punctate location with respect to kinetochore (NDC80) during ookinete development (Fig S2E).

### *Plasmodium* SAS4 is dispensable for parasite proliferation and transmission

Based on the expression and location of SAS4 during male gametogenesis and the essential role of BB protein SAS6 in male gametogenesis (Marques et al, 2015) we examined the importance of SAS4 in male gamete formation. We deleted the gene in a *P. berghei* line constitutively expressing GFP (WT-GFP, Fig S3A) (Janse et al, 2006). Diagnostic PCR showed successful integration of the targeting construct at the *sas4* locus (Fig S3B) and quantitative real time PCR (qRT-PCR) showed the lack of *sas4* expression in gametocytes, confirming the deletion of the *sas4* gene (Fig 4A). Successful creation of the Δ*SAS4* parasite indicated that the gene is not essential in asexual blood stages, consistent with the absence of the protein's expression at this stage of the life cycle in wild-type parasites. Further phenotypic analysis of the Δ*SAS4* parasite was carried in comparison with the parental parasite (WT-GFP).

First, we examined male gametogenesis, and surprisingly, we observed no significant difference in male gamete formation in the

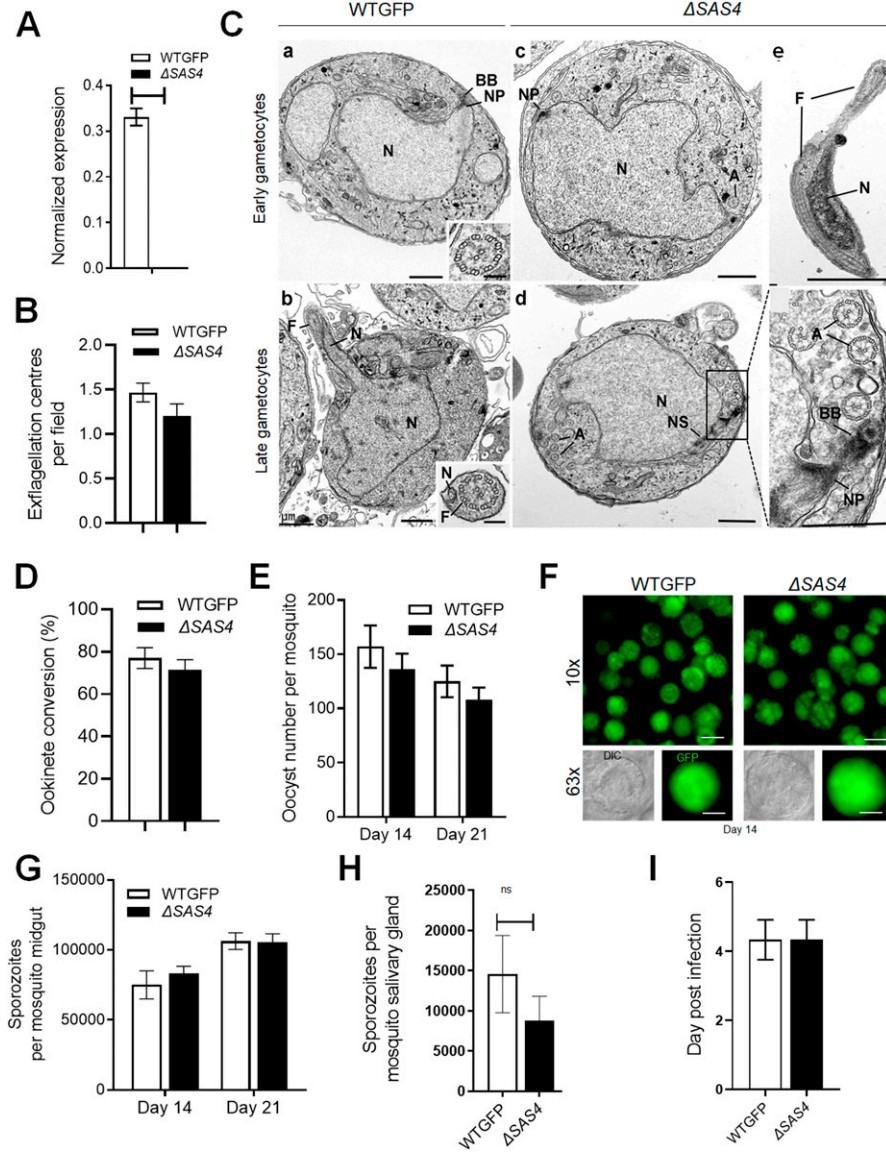

**Figure 4. SAS4 is dispensable for parasite proliferation and transmission.**
**(A)** qRT-PCR analysis of SAS4 transcript in the ΔSAS4 and WT-GFP parasites to show the complete depletion of *sas*4. **(B)** Male gametogenesis (exflagellation) of ΔSAS4 parasites compared with WT-GFP parasites measured as the number of exflagellation centres per field. Mean ± SEM. n = 3 independent experiments. **(C)** Electron micrographs of WT-GFP (a, b) and ΔSAS4 (c–e) male gametocytes. a. Early male gametocyte (4–5 minute post-activation) showing the large central nucleus with a nuclear pole (NP) and associated basal body. Bar is 1 μm. Insert. Cross section through an axoneme showing the normal 9 + 2 arrangement. Bar is 100 nm. (b) Late male gametocyte (15 minute post-activation) showing evidence of exflagellation with a portion of the nucleus (N) entering a cytoplasmic protrusion containing a flagellum (F). Insert: Cross section through a free male gamete showing the nucleus (N) and flagellum. Bar is 100 nm. (c) Early male gametocyte showing the large central nucleus with a nuclear pole (NP) and a few axonemes (A) in the cytoplasm. Bar is 1 μm. (d) Low power of a late-stage male gametocyte showing a nuclear spindle (NS) to one side of the nucleus (N) and a number of axonemes (A) within the cytoplasm. Bar is 1 μm. Enlargement of the enclosed area in (d) showing a nuclear pole (NP) with associated basal body. Note several axonemes (A) have the normal 9 + 2 microtubules. Bar is 1 μm. e. Part of a longitudinal section through a free male gamete showing the long flagellum (F) and adjacent nucleus (N). Bar is 1 μm. **(D)** Ookinete conversion as a percentage for ΔSAS4 and WT-GFP parasites. Ookinetes were identified using 13.1 antibody as a surface marker (P28) and defined as those cells that differentiated successfully into elongated "banana-shaped" ookinetes. Mean ± SEM. n = 3 independent experiments. **(E)** Total number of GFP-positive oocysts per infected mosquito in ΔSAS4 compared with WT-GFP parasites at 14 and 21-d post-infection. Mean ± SEM. n = 3 independent experiments. **(F)** Midguts at 10× and 63× magnification showing oocysts of ΔSAS4 and WT-GFP lines at 14 dpi. Scale bar = 50 μM in 10× and 20 μM in 63×. **(G)** Total number of sporozoites in oocysts of ΔSAS4 and WT-GFP parasites at 14 and 21 dpi. Mean ± SEM. n = 3 independent experiments. **(H)** Total number of sporozoites in salivary glands of ΔSAS4 and WT-GFP parasites. Bar diagram shows mean ± SEM. n = 3 independent experiments. ns, non-significant. **(I)** Bite back experiments showing successful transmission of WT-GFP and ΔSAS4 parasites from mosquito to mice. Mean ± SEM. n = 3 independent experiments.

Δ*sas*4 parasite in comparison with the WT-GFP parasite (Fig 4B). Speculating that there might be a defect in the subcellular structures, we performed in-depth analysis of WT-GFP and Δ*sas*4 gametocytes by EM. Electron micrographs of gametocytes activated for 4–5 min (early) showed no difference between Δ*sas*4 and WT-GFP parasites (Fig 4Ca and c). Similarly, there were no differences in electron micrographs of exflagellating gametocytes activated for 15 min (late) (Fig 4Cb, d, and e). Zygote formation and its differentiation to ookinete development were also unaffected (Fig 4D). To assess the effect of *sas*4 gene deletion on oocyst development, the number of GFP-positive oocysts on the mosquito gut wall was counted in mosquitoes fed with either ΔSAS4 or WT-GFP parasites; there was no significant difference in the number or size of ΔSAS4 oocysts compared with WT-GFP controls at 14- and 21-d post-infection (Fig 4E and F). The number of sporozoites produced by ΔSAS4 and WT-GFP parasites was comparable (Fig 4G), and although

there was a slight reduction in numbers of ΔSAS4 salivary gland sporozoites, the difference from WT-GFP numbers was not significant (Fig 4H). The infectivity of the ΔSAS4 sporozoites to naïve mice was similar to that of WT-GFP parasites (Fig 4I).

## Discussion

BBs are centriolar organelles that nucleate flagella and cilia, and are important MTOC components, with different and distinct ways of organisation that have arisen during centriole evolution in eukaryotes (Carvalho-Santos et al, 2011; Nabais et al, 2020). SAS6 and SAS4 are core protein components of this organelle (Carvalho-Santos et al, 2010; Hodges et al, 2010). *Plasmodium*, the evolutionarily divergent unicellular eukaryote and causative agent of malaria, shows a rapid and atypical process leading to formation of

flagellated male gametes within the mosquito gut, and a crucial transmission stage. Our recent and earlier ultrastructure studies had identified an amorphous BB in the cytoplasmic compartment of the male gametocyte, but the properties and function during unusual flagellum formation of SAS4, a conserved CBB molecule, were unknown (Sinden et al, 1976; Zeeshan et al, 2019a). During the *Plasmodium* life cycle, a centriole is only present during male gametogenesis, whereas during mitosis in other proliferative stages only the amorphous acentriolar MTOC is present (Sinden, 1991). Here we have investigated by live cell imaging in real time the profile of SAS4/CPAP to understand whether it is involved in axoneme biogenesis during male gamete formation and how its replication is coordinated during mitosis.

Male gametocyte activation results in rapid genome replication from 1N to 8N in 8 min, with three rounds of mitosis without nuclear division (Sinden, 1991). Our imaging suggests a very rapid de novo formation of SAS4 during male gametogenesis, with a dynamic profile in the cytoplasm. The number of discrete SAS4-GFP foci duplicates as genome replication and rounds of mitosis occur. In most cells, SAS4 appears to coalesce into a close doublet at the beginning of the first mitosis and then this structure replicates in coordination with replication of the genome. We show that eight BB-like structures, each with a close SAS4 doublet are formed de novo and are present at the end of genome replication and mitosis. These SAS4 foci appear to be in the cytoplasm of the cell and associated with the outside of the nucleus, suggesting that SAS4 is part of the BB of the outer MTOC. To confirm its location, we generated parasite lines expressing both SAS4-GFP and either the cytoplasmic BB and axoneme marker, kinesin-8B-mCherry (Zeeshan et al, 2019a), or NDC80-mCherry, a kinetochore marker of the mitotic spindle in the nucleus (Zeeshan et al, 2020b). Real time imaging clearly delineated the spatial organisation of SAS4 with respect to these complementary cytoplasmic and nuclear markers. It was clear that SAS4 is part of the BB MTOC structure with a similar spatial profile to that of kinesin-8B. However, SAS4 and kinesin-8B do not colocalise and the images suggest that SAS4 may be located at the centre of the BB that nucleates axoneme assembly during the male cell differentiation. Whereas kinesin-8B is part of the axoneme assembly, SAS4 is limited to the BB. This is consistent with previous results showing SAS4 in cytoplasmic compartment associated with the dense BB (Rashpa & Brochet, 2022). We show that SAS4 duplication is synchronized with the accumulation of NDC80 and spindle formation during successive rounds of genome replication during mitosis. However, as we have shown previously, the NDC80 foci are within the nucleus, and our analysis suggests that SAS4 is likely a component of the BB/MTOC in the outer cytoplasmic compartment. We suggest that the MTOC/spindle assembly marked with NDC80 inside the nucleus is coupled together with the cytoplasmic BB/MTOC as cytoskeletal structures. Although *Plasmodium* undergoes closed mitosis, it is possible that these two components of the cell can coordinate mitosis and axoneme assembly to ensure that there is one flagellum for each genome/nucleus at exflagellation. These two components may be inter-connected as part of a bipartite MTOC. A similar structure appears to be involved during both sexual and asexual (tachyzoites) replication of another apicomplexan parasite, *Toxoplasma gondii* (Suvorova et al, 2015). However, this centriole only acts as a

BB MTOC giving rise axoneme formation during formation of the bi-flagellated male gamete during male gametogenesis (Ferguson et al, 1974). Morphologically, male gametes of Toxoplasma and Eimeria have similar BB structures (Ferguson et al, 1974, 1977), although the location of associated molecules in unknown as no suitable in vitro system for the study of male gametogenesis is available for the Coccidia. In trypanosomes the BB is coupled with kinetoplast DNA during cell division (Vaughan & Gull, 2015) and SAS4 controls the cell cycle transition (Hu et al, 2015), suggesting that the evolution of BBs has depended upon the requirement of the cell to multiply in different niches. Axoneme and flagellum formation only occur during male gametogenesis in *Plasmodium*, and their absence during blood stage schizogony is mirrored by the lack of SAS4 expression at this stage of the life cycle.

We next examined the functional role of SAS4 in *Plasmodium* by examining the phenotype resulting from gene deletion. In contrast to *Plasmodium* SAS6 that was shown to be important for male gamete formation (Marques et al, 2015), we found that SAS4 is not essential for exflagellation as there was no significant change in flagellated gamete formation. Our findings are corroborated with a recent study showing a non-essential role of SAS4 in parasite transmission, although it described a significant decrease in male gamete formation (Rashpa & Brochet, 2022). At other stages of parasite development, including zygote formation, ookinete and sporozoite development, and parasite transmission and infectivity, no significant differences from the WT-GFP parasite were observed. We conclude that the presence of SAS4 is not essential for parasite development throughout the life cycle. It is possible that other proteins compensate for SAS4 function, or it has a redundant function in *Plasmodium*.

It will be interesting in the future to analyse in depth the 3D structure of *Plasmodium* BBs and their components, for example, by using these BB and mitotic spindle markers and correlative light and electron microscopy (CLEM) as described recently for the bryophyte *Physcomitrium* (Pereira et al, 2021). In a recent study, Raspa and Brochet have used expansion microscopy to study the MTOC structures showing its bipartite architecture in *Plasmodium* (Rashpa & Brochet, 2022), complementing our findings by live cell imaging, super resolution, and EM. The protein interactome obtained using these lines and others will provide tools to identify BB components and understand their evolution in this divergent organism, which assembles the complete BB complement de novo in 8 min. This is extremely fast, for example, when compared with the rapid assembly that takes 1 h in Naegleria (Fritz-Laylin et al, 2016). These approaches may also identify the conserved and divergent molecules that enable the extremely fast flagellum assembly, which is one of the fastest known and where accuracy may be compromised because of the need for speed (Sinden et al, 2010; Fritz-Laylin et al, 2016).

Overall, this study shows that SAS4 is part of an outer cytoplasmic BB MTOC where there is a need for coordination between flagellum assembly in the cytoplasm and genome replication in the nucleus so that there is one flagellum for each haploid nucleus formed following karyokinesis. Formation of the BB occurs de novo and the entire process is very rapid. However, the deletion of the *SAS4* gene does not affect male gametogenesis and the gene is not essential for parasite transmission or at other stages of the life cycle.

# Materials and Methods

## Ethics statement

The animal work passed an ethical review process and was approved by the United Kingdom Home Office. Work was carried out under UK Home Office Project Licenses (30/3248 and PDD2D5182) in accordance with the United Kingdom "Animals (Scientific Procedures) Act 1986." 6–8-wk-old female CD1 outbred mice from Charles River laboratories were used for all experiments.

## Generation of transgenic parasites

The C-terminus of SAS4 was tagged with GFP by single crossover homologous recombination in the parasite. To generate the SAS4-GFP line, a region of the *sas4* gene downstream of the ATG start codon was amplified using primers T2011 and T2012, ligated to p277 vector, and transfected as described previously (Saini et al, 2017). A schematic representation of the endogenous *sas4* locus (PBANKA_1322200), the constructs and the recombined *sas4* locus are shown in Fig S1B. The oligonucleotides used to generate the mutant parasite lines are described in Table S1. *P. berghei* ANKA line 2.34 (for GFP tagging) or ANKA line 507cl1 expressing GFP (for gene deletion) were transfected by electroporation (Janse et al, 2006).

The gene-deletion targeting vector for *sas4* was constructed using the pBS-DHFR plasmid, which contains polylinker sites flanking a *T. gondii dhfr/ts* expression cassette conferring resistance to pyrimethamine, as described previously (Zeeshan et al, 2019a). PCR primers N1391 and N1392 were used to generate an 803 bp fragment of *sas4* 5′ upstream sequence from genomic DNA, which was inserted into *Apa*I and *Hin*dIII restriction sites upstream of the dhfr/ts cassette of pBS-DHFR. A 721 bp fragment generated with primers N1393 and N1394 from the 3′ flanking region of *sas4* was then inserted downstream of the dhfr/ts cassette using *Eco*RI and *Xba*I restriction sites. The linear targeting sequence was released using *Apa*I/*Xba*I. A schematic representation of the endogenous *sas4* locus the constructs and the recombined *sas4* locus can be found in Fig S3A.

## Parasite genotype analyses

For the parasites expressing a C-terminal GFP-tagged SAS4 protein, diagnostic PCR was used with primer 1 (IntT201) and primer 2 (ol492) to confirm integration of the GFP targeting construct (Fig S1C). For the gene KO parasites, diagnostic PCR was used with primer 1 (IntN139) and primer 2 (ol248) to confirm integration of the targeting construct, and primer 3 (N139 KO1) and primer 4 (N139 KO2) were used to confirm deletion of the *sas4* gene (Fig S3B).

## Purification of gametocytes

The purification of gametocytes was achieved using a protocol described previously (Beetsma et al, 1998) with some modifications. Briefly, parasites were injected into phenylhydrazine-treated mice and enriched by sulfadiazine treatment after 2 d of infection. The blood was collected on day 4 after infection and gametocyte-infected cells were purified on a 48% vol/vol NycoDenz (in PBS) gradient (NycoDenz stock solution: 27.6% wt/vol NycoDenz in 5 mM Tris–HCl, pH 7.20, 3 mM KCl, and 0.3 mM EDTA). The gametocytes were harvested from the interface and washed.

## Live cell- and time-lapse imaging

Different developmental stages of parasites during schizogony, gametogenesis and zygote to ookinete transformation were analyzed for SAS4-GFP expression and localization using a 63× oil immersion objective on a Zeiss AxioImager M2 microscope. Purified male gametocytes were examined for GFP expression and localization at different time points (1–15 min) after activation in ookinete medium containing xanthurenic acid. Images were captured using a 63× oil immersion objective on a Zeiss AxioImager M2 microscope fitted with an AxioCam ICc1 digital camera (Carl Zeiss, Inc.). Time-lapse videos (1 frame every 5 s for 10 cycles) were taken with a 63× objective lens on the same microscope and analyzed with the AxioVision 4.8.2 software as described recently (Zeeshan et al, 2020b).

## Generation of dual-tagged parasite lines

The SAS4-GFP parasites were mixed with kinesin-8B-mCherry or NDC80-mCherry parasites in equal numbers and injected into a mouse. Mosquitoes were fed on this mouse 4–5 d after infection when gametocyte parasitaemia was high. These mosquitoes were checked for oocyst development and sporozoite formation at day 14 and day 21 after feeding. Infected mosquitoes were then allowed to feed on naïve mice and after 4–5 d these mice were examined for blood stage parasitaemia by microscopy with Giemsa-stained blood smears. In this way, some parasites expressed both SAS4-GFP and kinesin-8B-mCherry or SAS4-GFP and NDC80-mCherry in the resultant gametocytes. These gametocytes were purified, and fluorescence microscopy images were collected as described above.

## Immunofluorescence assay

The purified SAS4-GFP gametocytes were activated in ookinete medium then fixed at 2–8 min post-activation with 4% PFA diluted in microtubule stabilizing buffer (MTSB) for 10–15 min and added to poly-L-lysine coated eight-well slides. Immunocytochemistry was performed using primary GFP-specific rabbit monoclonal antibody (mAb) (A1122; used at 1:250; Invitrogen) and primary mouse anti–*α* tubulin mAb (T9026; used at 1:1,000; Sigma-Aldrich). Secondary antibodies were Alexa 488–conjugated anti-mouse IgG (A11004; Invitrogen) and Alexa 568–conjugated anti-rabbit IgG (A11034; Invitrogen) (used at 1 in 1,000). The slides were then mounted in VECTASHIELD 19 with DAPI (Vector Labs) for fluorescence microscopy.

## Super resolution microscopy

A small volume (3 *μ*l) of gametocytes was mixed with Hoechst dye and pipetted onto 2% agarose pads (5 × 5 mm squares) at room temperature. After 3 min, these agarose pads were placed onto

glass-bottom dishes with the cells facing towards the glass surface (P35G-1.5-20-C; MatTek). Cells were scanned with an inverted microscope using Zeiss C-Apochromat 63×/1.2 W Korr M27 water immersion objective on a Zeiss Elyra PS.1 microscope, using the SIM technique. The correction collar of the objective was set to 0.17 for optimum contrast. The following settings were used in SIM mode: lasers, 405 nm: 20%, 488 nm: 16%; 561 nm: 8%, exposure; times 200 ms (Hoechst) 100 ms (GFP); 200 ms (mCherry), five grid rotations, five phases. The band pass filters BP 420–480 + LP 750, BP 495–550 + LP 750 and BP570-620 + LP 750 were used for the blue, green and red channels, respectively. Multiple focal planes (Z stacks) were recorded with 0.2-$\mu$m step size; later post-processing, a Z correction was done digitally on the 3D rendered images to reduce the effect of spherical aberration (reducing the elongated view in Z; a process previously tested with fluorescent beads). Registration (channel alignment) correction was applied based on fluorescent beads images. Images were processed and all focal planes were digitally merged into a single plane (Maximum intensity projection). The images recorded in multiple focal planes (Z stack) were 3D rendered into virtual models and exported as images. Processing and export of images were done by Zeiss Zen 2012 Black edition, Service Pack 5 and Zeiss Zen 2.1 Blue edition (Zeeshan et al, 2020b).

### Parasite phenotype analyses

Blood containing ~50,000 parasites of the ΔSAS44 line was injected i.p. into mice to initiate infections. Asexual stages and gametocyte production were monitored by microscopy on Giemsa-stained thin smears. 4–5 d post-infection, exflagellation and ookinete conversion were examined as described previously (Zeeshan et al, 2019b) with a Zeiss AxioImager M2 microscope (Carl Zeiss, Inc.) fitted with an AxioCam ICc1 digital camera. To analyse mosquito transmission, 30–50 Anopheles stephensi SD 500 mosquitoes were allowed to feed for 20 min on anaesthetized, infected mice with an asexual parasitaemia of 15% and a comparable number of gametocytes as determined on Giemsa-stained blood films. To assess midgut infection, ~15 guts were dissected from mosquitoes on day 14 post-feeding, and oocysts were counted on an AxioCam ICc1 digital camera fitted to a Zeiss AxioImager M2 microscope using a 63× oil immersion objective. On day 21 post-feeding, another 20 mosquitoes were dissected, and their guts crushed in a loosely fitting homogenizer to release sporozoites, which were then quantified using a haemocytometer. Mosquito bite back experiments were performed 21 d post-feeding using naive mice, and blood smears were examined after 3–4 d.

### EM

Gametocytes activated for 4–5 and 15 min were fixed in 4% glutaraldehyde in 0.1 M phosphate buffer and processed for EM as previously described (Zeeshan et al, 2019b). Briefly, samples were post fixed in osmium tetroxide, treated en bloc with uranyl acetate, dehydrated, and embedded in Spurr's epoxy resin. Thin sections were stained with uranyl acetate and lead citrate before examination in a JEOL1200EX electron microscope (Jeol UK Ltd).

### Quantitative real time PCR (qRT-PCR) analyses

RNA was isolated from gametocytes using an RNA purification kit (Stratagene). cDNA was synthesised using an RNA-to-cDNA kit (Applied Biosystems). Gene expression was quantified from 80 ng of total RNA using a SYBR green fast master mix kit (Applied Biosystems). All the primers were designed using the primer3 software (Primer-blast, NCBI). Analysis was conducted using an Applied Biosystems 7500 fast machine with the following cycling conditions: 95°C for 20 s followed by 40 cycles of 95°C for 3 s; 60°C for 30 s. Three technical replicates and three biological replicates were performed for each assayed gene. The hsp70 (PBANKA_0818900) and arginyl-t RNA synthetase (PBANKA_1434200) genes were used as endogenous control reference genes. The primers used for qPCR can be found in Table S1.

### Statistical analysis

All statistical analyses were performed using GraphPad Prism 7 (GraphPad Software). For qRT-PCR, an unpaired t test was used to examine significant differences between wild-type and mutant strains.

## Supplementary Information

## Acknowledgements

We thank the Oxford Brookes Centre for Bioimaging for assistance. This work was supported by: Medical Research Council, UK (G0900278, MR/K011782/1) and Biotechnology and Biological Sciences Research Council (BBSRC) (BB/N017609/1) to R Tewari and M Zeeshan; the Francis Crick Institute (FC001097), the Cancer Research UK (FC001097), the UK Medical Research Council (FC001097), and the Wellcome Trust (FC001097) to AA Holder. The super resolution microscope facility was funded by the BBSRC grant BB/L013827/1. For the purpose of Open Access, the author has applied a CC BY public copyright licence to any Author Accepted Manuscript version arising from this submission.

### Author Contributions

M Zeeshan: formal analysis, validation, investigation, visualization, methodology, and writing—original draft, review, and editing.
D Brady: validation, investigation, and methodology.
R Markus: investigation, visualization, and methodology.
S Vaughan: resources and software.
D Ferguson: conceptualization, supervision, investigation, visualization, methodology, and writing—original draft, review, and editing.
AA Holder: conceptualization, supervision, funding acquisition, and writing—review and editing.
R Tewari: conceptualization, resources, data curation, formal analysis, supervision, funding acquisition, validation, investigation,

visualization, methodology, and writing—original draft, review, and editing.

## Conflict of Interest Statement

The authors declare that the research was conducted in the absence of any commercial or financial relationships that could be construed as a potential conflict of interest.

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
