## [Reviewer comments · Life Science Alliance]

Life Science Alliance

Plasmodium SAS4: basal body component of male cell which is dispensable for parasite transmission

Mohammad Zeeshan, Declan Brady, Robert Markus, Sue Vaughan, David Ferguson, Tony Holder, and Rita Tewari
DOI: <https://doi.org/10.26508/lsa.202101329>

Corresponding author(s): Rita Tewari, University of Nottingham

Review Timeline:	Submission Date:	2021-12-06
	Editorial Decision:	2022-01-03
	Revision Received:	2022-04-07
	Editorial Decision:	2022-04-21
	Revision Received:	2022-04-28
	Accepted:	2022-04-28

Scientific Editor: Novella Guidi

Transaction Report:

January 3, 2022

Re: Life Science Alliance manuscript #LSA-2021-01329

Prof. Rita Tewari
University of Nottingham
School of Life Sciences
Queens Medical Centre
Nottingham NG7 2UH
United Kingdom

Dear Dr. Tewari,

Thank you for submitting your manuscript entitled "Plasmodium SAS4/CPAP is a flagellum basal body component during male gametogenesis but is not essential for parasite transmission" to Life Science Alliance. The manuscript was assessed by expert reviewers, whose comments are appended to this letter. We, thus, encourage you to submit a revised version of the manuscript back to LSA that responds to all of the reviewers' points.

Thank you for this interesting contribution to Life Science Alliance. We are looking forward to receiving your revised manuscript.

Sincerely,

B. MANUSCRIPT ORGANIZATION AND FORMATTING:

Reviewer #1 (Comments to the Authors (Required)):

This manuscript by Tewari and colleagues examines the organization of a protein called SAS4/CPAP, which is a component of the basal body (BB) in *Plasmodium berghei*. SAS4/CPAP is expressed in late stage male gametocytes and is associated with the microtubule organising centre (MTOC) that nucleates the gamete flagella.

The work provides an important clarification of the roles of the centriole-containing BB MTOC in the cytoplasm of the male gametocytes and the acentriolar plaque MTOC that is embedded in the nuclear membrane and which nucleates the spindle microtubules.

The work is detailed; and the findings clearly explained.

One aspect that would benefit from further work is the fact that PlasmoDB data suggests that SAS4 transcript are present in asexual schizonts as well as gametocytes

(https://plasmodb.org/plasmo/app/record/gene/PBANKA_1322200#ExpressionGraphs).

Here, the authors used fluorescence microscopy to follow the expression and report that SAS4-GFP is observed as punctate structures, only in male gametocytes. While separation of *P. berghei* asexual and gametocyte stages is not trivial, it would be useful to provide Western analysis to confirm that the tagged protein has the correct size and help confirm that expression is limited to activated male gametocytes. Western analysis would also confirm that the knockout is successful.

Minor points

Line 188. The authors state that "SAS4 is associated with kinesin-8B during a very early stage of basal body formation and remains associated with it throughout axoneme assembly during the rest of male gametogenesis". That statement suggests a close physical association. In fact, the super-res microscopy data indicate that SAS4 and kinesin-8B are located in adjacent structures. The association of the two markers with distinct, though connected structures, could be made clearer.

In the diagram in Fig 2C, the NDC80 structure appears to be outside the nucleus in some of the panels. This could be adjusted.

Line 230. "The outer mass is a typical basal body with nine single α -tubules." The data presented do not permit quantification of the number of microtubules. I assume the authors means that the observed structure has an ultrastructural appearance that is consistent with that of a typical basal body.

Reviewer #2 (Comments to the Authors (Required)):

1. This is an excellent piece of work giving insight into and involvement of a protein called SAS4, in the coordination of two microtubule organizing centres in the process of malaria parasite male gamete formation. Here it is necessary for three rapid nuclear divisions to be coordinated with the generation of 8 axonemes in the cytoplasm on each of the resulting flagellated male gametes.

2. The study is very well carried out using reverse genetics, epitope tagging and beautiful high quality imaging. I believe that the results have all been well interpreted and explained using very helpful schematics.

3. I have no major issues at all with the manuscript and have found only a very small number of minor issues/errors. The images in Figure 2 are high quality and the findings are clear aided by the schematic. However, a slight issue is the use of different colours for kinesin 8B and for NDC80. This is a little confusing especially when the schematic in Figure 2B chooses one of these two alternative colours. For example, kinesin 8B is magenta in the schematic in Figure 2B and the right panels of Figures 2A and 2C, but is red in the left panel of Figure A. So the colour scheme of the schematic does not correspond to the colour scheme of the left panel of Figure 2A. The same red/magenta problem occurs for the colour of NDC80 in the right panel of Figure 2C. As it stands I think it is a bit confusing for the reader. I'm not sure what the best way to solve this problem is, but

one possibility is to use a completely different colour scheme in Figure 2B schematic (?). If the authors could give some thought to dealing with this small issue it would be great. The figure legend does not help as it refers only to the two data panels (labelled A and B) and does not mention the schematic and so this needs to be corrected. The actual figures are not numbered in the merged manuscript/figures pdf file, but fortunately it is obvious which figure is which.

Line 188 Change to 'these data show'.

Line 222 I would change 'doing similar organisation' to e.g. 'performing the same role'

Line 228 Perhaps say 'our micrographs' to clarify that the new images, rather than the Sinden (1976) images, are being referred to.

Line 237, delete full stop after MTs.

Line 256, delete 'known as exflagellation' as this is stated earlier. I would also replace 'flagellate gamete' with 'male gamete' (or 'flagellated gamete').

Line 273 is lacking a full stop. Alternatively insert 'and' before SAS6 and SAS4.

Line 277-81 lacks references on the authors' previous work that is mentioned.

Line 298 has an incomplete sentence: 'basal body outer MTOC of.'

Line 321 I think it would be better to specify the Toxoplasma life cycle stage here rather than just 'asexual cells'. Also male gametes (plural) and 'have' to replace 'has'.

Line 347 lacks a full stop.

Reviewer #3 (Comments to the Authors (Required)):

The paper examines the localisation of SAS4, a basal body associated protein in Plasmodium, where flagella are built in a non-canonical way. Using different labels for proteins associated with spindle microtubules and axonemal microtubules the authors can show that formation of these microtubule complexes are coupled, although data from the 1970 already suggested this. Gene deletion did not reveal a phenotype which limits the interest of the study but the images are nice and the conclusion largely warranted if not entirely new. Please consider the following points during revision or submission elsewhere:

Please show SAS4 localization in free gametes.

Throughout the manuscript: should Kinesin not be written kinesin?

30, maybe introduce the name 'centriolar plaque' that is given to the Plasmodium MTOC here?

56, 'The canonical view... is associated...' maybe rephrase?

65, 'occurs' instead of 'is'

84, 'appear'

107-109: not sure I would describe the two MTOCs (as described previously as two) as a single MTOC with two parts. I would rather stay with the observation/suggestion that two MTOCs are linked (or fused).

127, this sentence of conclusion about SAS4 dynamics after saying that a KO was generated is somewhat confusing, maybe move the sentence up by one sentence

137, introduce *P. berghei* as a model rodent infecting parasite

154, I only see two dots that look bulged like 'dimers' in the image, no tetrads as in the model in Figure 1A, please adjust or comment on prior data from EM that shows four units. Maybe also rotate the cartoon (guide) so that it fits in orientation to the images shown in the row directly below.

163, now I see the tetrads - maybe refer to this already before.

Figure 2A: please adjust writing to on line for kinesin-8B

For figure 2 it would be really nice to see microtubules in addition to kinesin-8B and NDC80 staining. Maybe add the images to figure 1 using a 'red' anti-tubulin antibody?

Figure 3: Maybe use 'nuclear pole' instead of NP, as many readers will think NP as nuclear pore.

In Figure 2B: maybe indicate the nuclear envelope, I guess the large red line (at minute 3) is within the nucleus.

298, sentence not complete

Reviewers' comments with responses (In red)

We thank the reviewers for their positive and constructive comments and their suggestions to improve the manuscript, which we have carefully addressed. We hope that these revisions satisfy all the reviewers' queries and concerns. We provide below our responses (highlighted in red) to specific points made by the reviewers:

Reviewer #1 (Comments to the Authors (Required)):

This manuscript by Tewari and colleagues examines the organization of a protein called SAS4/CPAP, which is a component of the basal body (BB) in *Plasmodium berghei*. SAS4/CPAP is expressed in late stage male gametocytes and is associated with the microtubule organising centre (MTOC) that nucleates the gamete flagella.

The work provides an important clarification of the roles of the centriole-containing BB MTOC in the cytoplasm of the male gametocytes and the acentriolar plaque MTOC that is embedded in the nuclear membrane and which nucleates the spindle microtubules. The work is detailed; and the findings clearly explained.

Authors' response: We thank the reviewer for recognising our work as detailed and clearly explained findings.

One aspect that would benefit from further work is the fact that PlasmoDB data suggests that SAS4 transcript are present in asexual schizonts as well as gametocytes (https://plasmodb.org/plasmo/app/record/gene/PBANKA_1322200#ExpressionGraphs). Here, the authors used fluorescence microscopy to follow the expression and report that SAS4-GFP is observed as punctate structures, only in male gametocytes.

Authors' response: We thank the reviewer for highlighting the PlasmoDB data showing the presence of SAS4 transcript in asexual blood stage schizonts. Consistently, our qPCR data also identified SAS4 transcripts in schizonts (please see Fig S1A). However, we did not detect SAS4-GFP fusion protein in asexual blood stages by live cell imaging (please see Fig S2A), unlike in the male gametocytes where punctate structures of SAS4-GFP were visible in the cytoplasm (please see Fig S2B). We have added a few sentences in the result section between line numbers 138-142).

While separation of *P. berghei* asexual and gametocyte stages is not trivial, it would be useful to provide Western analysis to confirm that the tagged protein has the correct size and help confirm that expression is limited to activated male gametocytes.

Authors' response: We have now done western analysis confirming the correct size of the fusion protein (SAS4-GFP) and that its expression is limited to gametocytes (Fig S1C).

Western analysis would also confirm that the knockout is successful.

Authors' response: We agree with reviewer that a western blot with antibody against SAS4 would confirm the deletion of the gene. However, we could not perform this experiment because we do not have such antibodies, and instead we confirmed the gene deletion by qPCR analysis (Fig 4A).

Minor points

Line 188. The authors state that "SAS4 is associated with kinesin-8B during a very early stage of basal body formation and remains associated with it throughout axoneme assembly during the rest of male gametogenesis". That statement suggests a close physical association. In fact, the super-res microscopy data indicate that SAS4 and kinesin-8B are located in adjacent structures. The association of the two markers with distinct, though connected structures, could be made clearer.

Authors' response: We have modified the sentence and made clearer their distinct locations in line numbers 203.

In the diagram in Fig 2C, the NDC80 structure appears to be outside the nucleus in some of the panels. This could be adjusted.

Authors' response: We have tried to adjust the nuclear staining. We have also provided additional images showing prominent NDC80 and nuclear signals (please see Fig S2C).

Line 230. "The outer mass is a typical basal body with nine single α -tubules." The data presented do not permit quantification of the number of microtubules. I assume the authors means that the observed structure has an ultrastructural appearance that is consistent with that of a typical basal body.

Authors' response: We agree with the reviewer. The outer mass has a typical basal body structure. The nine single α -tubules are not very clear in electron micrographs, but its ultrastructural appearance is consistent with that of a typical basal body. Now we have provided two more electron micrographs (please see Fig 3B).

Reviewer #2 (Comments to the Authors (Required)):

1. This is an excellent piece of work giving insight into and involvement of a protein called SAS4, in the coordination of two microtubule organizing centres in the process of malaria parasite male gamete formation. Here it is necessary for three rapid nuclear divisions to be coordinated with the generation of 8 axonemes in the cytoplasm on each of the resulting flagellated male gametes.

Authors' response: We thank the reviewer for considering it as an excellent piece of work giving insight into and involvement of SAS4.

2. The study is very well carried out using reverse genetics, epitope tagging and beautiful high-quality imaging. I believe that the results have all been well interpreted and explained using very helpful schematics.

Authors' response: We thank again the reviewer for believing our results are well interpreted and explained using very helpful schematics.

3. I have no major issues at all with the manuscript and have found only a very small number or minor issues/errors.

The images in Figure 2 are high quality and the findings are clear aided by the schematic. However, a slight issue is the use of different colours for kinesin 8B and for NDC80. This is a little confusing especially when the schematic in Figure 2B chooses one of these two alternative colours. For example, kinesin 8B is magenta in the schematic in Figure 2B and the right panels of Figures 2A and 2C, but is red in the left panel of Figure A. So the colour scheme of the schematic does not correspond to the colour scheme of the left panel of Figure 2A.

Authors' response: We are sorry for this confusion. We have now used exactly the same colours in the schematic (red for kinesin-8B and green for SAS4) shown in left panel. We also want to make it clear that the schematic diagram corresponds to the left panel of Fig 2A only.

The same red/magenta problem occurs for the colour of NDC80 in the right panel of Figure 2C. As it stands I think it is a bit confusing for the reader. I'm not sure what the best way to solve this problem is, but one possibility is to use a completely different colour scheme in Figure 2B schematic (?). If the authors could give some thought to dealing with this small issue it would be great.

Authors' response: We have now used the same colours in the schematic (red for NDC80 and green for SAS4) that correspond to the left panel of Fig 2B only. We have removed the combined schematic to avoid the confusion.

The figure legend does not help as it refers only to the two data panels (labelled A and B) and does not mention the schematic and so this needs to be corrected. The actual figures are not numbered in the merged manuscript/figures pdf file, but fortunately it is obvious which figure is which.

Authors' response: As we have removed the combined schematic, now the legends correspond to the current Figs 2A and 2B.

Line 188 Change to 'these data show'.

Authors' response: We have made the change at line 203.

Line 222 I would change 'doing similar organisation' to e.g. 'performing the same role'

Authors' response: Changed. Now at line number 242.

Line 228 Perhaps say 'our micrographs' to clarify that the new images, rather than the Sinden (1976) images, are being referred to.

Authors' response: Mentioned as our micrographs. Now at line number 228.

Line 237, delete full stop after MTs.

Authors' response: Deleted. Line 247

Line 256, delete 'known as exflagellation' as this is stated earlier. I would also replace 'flagellate gamete' with 'male gamete' (or 'flagellated gamete').

Authors' response: Deleted and replaced with male gamete. Now at line 293.

Line 273 is lacking a full stop. Alternatively insert 'and' before SAS6 and SAS4.

Authors' response: Inserted full stop. Line 294

Line 277-81 lacks references on the authors' previous work that is mentioned.

Authors' response: Added the references. Line 322

Line 298 has an incomplete sentence: 'basal body outer MTOC of.'

Authors' response: Corrected, line 340.

Line 321 I think it would be better to specify the Toxoplasma life cycle stage here rather than just 'asexual cells'. Also male gametes (plural) and 'have' to replace 'has'.

Authors' response: Mentioned asexual cells as tachyzoites. Also replaced with gametes (plural) and have. Now at line 362 and 367.

Line 347 lacks a full stop.

Authors' response: Inserted full stop. Line 387.

We thank the reviewer for pointing out these corrections/suggestions

Reviewer #3 (Comments to the Authors (Required)):

The paper examines the localisation of SAS4, a basal body associated protein in Plasmodium, where flagella are built in a non-canonical way. Using different labels for proteins associated with spindle microtubules and axonemal microtubules the authors can show that formation of these microtubule complexes are coupled, although data from the 1970 already suggested this. Gene deletion did not reveal a phenotype which limits the

interest of the study but the images are nice and the conclusion largely warranted if not entirely new. Please consider the following points during revision or submission elsewhere:

Please show SAS4 localization in free gametes.

Authors' response: Now we have shown the SAS4 localization in free gametes (Fig 1A).

Throughout the manuscript: should Kinesin not be written kinesin?

Authors' response: Now written as kinesin throughout the manuscript.

30, maybe introduce the name ,centriolar plaque' that is given to the Plasmodium MTOC here?

Authors' response: Thanks for the suggestion. Now we have Introduced this at line 31.

56, 'The canonical view... is associated...' maybe rephrase?

Authors' response: Thanks for the suggestions. We have now rephrased it at line 58.

65, 'occurs' instead of 'is'

Authors' response: Corrected at line 67.

84, 'appear'

Authors' response: corrected at line 85

107-109: not sure I would describe the two MTOCs (as described previously as two) as a single MTOC with two parts. I would rather stay with the observation/suggestion that two MTOCs are linked (or fused).

Authors' response: Modified as suggested in line numbers 109-111.

127, this sentence of conclusion about SAS4 dynamics after saying that a KO was generated is somewhat confusing, maybe move the sentence up by one sentence

Authors' response: moved at line number 118.

137, introduce P. berghei as a model rodent infecting parasite

Authors' response: Thanks for the suggestion. Now we have introduced it at line number 144.

154, I only see two dots that look bulged like 'dimers' in the image, no tetrads as in the model in Figure 1A, please adjust or comment on prior data from EM that shows four units. Maybe also rotate the cartoon (guide) so that it fits in orientation to the images shown in the row directly below.

163, now I see the tetrads - maybe refer to this already before.

Authors' response: Thanks to reviewer for the suggestions. We have now rearranged Figure 1, refer to Fig 1B in the text and have changed the orientation of cartoon.

Figure 2A: please adjust writing to on line for kinesin-8B

Authors' response: Thanks. Adjusted

For figure 2 it would be really nice to see microtubules in addition to kinesin-8B and NDC80 staining. Maybe add the images to figure 1 using a 'red' anti-tubulin antibody?

Authors' response: We thank the reviewer for this suggestion. We have now included images showing microtubules acquired from immunofluorescence assays using anti-tubulin antibody. Please see new fig 2C.

Figure 3: Maybe use 'nuclear pole' instead of NP, as many readers will think NP as nuclear pore.

Authors' response: Thanks for the suggestion. We have included the full forms in Figure 3.

In Figure 2B: maybe indicate the nuclear envelope, I guess the large red line (at minute 3) is within the nucleus.

Authors' response: We have now indicated the nuclear envelope in the schematic and described it separately in Fig 2A and 2B.

298, sentence not complete

Authors' response: Thanks to reviewer for pointing out this mistake. We have now corrected the sentence that is now at line number 340.

April 21, 2022

RE: Life Science Alliance Manuscript #LSA-2021-01329R

Prof. Rita Tewari
University of Nottingham
School of Life Sciences
Queens Medical Centre
Nottingham NG7 2UH
United Kingdom

Dear Dr. Tewari,

Thank you for submitting your revised manuscript entitled "Plasmodium SAS4: basal body component of male cell which is dispensable for parasite transmission". We would be happy to publish your paper in Life Science Alliance pending final revisions necessary to meet our formatting guidelines.

- please upload your manuscript text in doc format
- please add author contributions to your main manuscript text
- please add a conflict of interest statement to your main manuscript text
- please use the [10 author names, et al.] format in your references (i.e. limit the author names to the first 10)

Figure Issues:

- Figure 3A: The top part of the zoomed in part seems to be cut off and doesn't perfectly match the inset. Please provide the appropriate zoomed-in part.
- Figure 4C: you need boxes to indicate where the zoomed in parts are coming from. For example, C-e looks like it matches the inset of C-d, but this is not very clear in the figure. Also, what part of the figure C-f is from? (I.e. C-f is a zoomed in part of which figure?)
- Figure 4F: you should indicate with boxes to show which part of the main image is then zoomed in

A. FINAL FILES:

B. MANUSCRIPT ORGANIZATION AND FORMATTING:

Sincerely,

Reviewer #1 (Comments to the Authors (Required)):

This manuscript by Tewari and colleagues examines the organization of a protein called SAS4/CPAP, which is a component of the basal body (BB) in *Plasmodium berghei*. SAS4/CPAP is expressed in late stage male gametocytes and is associated with the microtubule organising centre (MTOC) that nucleates the gamete flagella.

The work provides an important clarification of the roles of the centriole-containing BB MTOC in the cytoplasm of the male gametocytes and the acentriolar plaque MTOC that is embedded in the nuclear membrane and which nucleates the spindle microtubules.

The authors have addressed my queries from the original submission.

Authors' responses (In blue)

RE: Life Science Alliance Manuscript #LSA-2021-01329R

We thank the editor for positive and constructive comments and suggestions to improve the figures, which we have carefully addressed, and inserted the additional information needed. We provide below our responses (highlighted in blue) to specific points made by the editor:

-please upload your manuscript text in doc format

We have uploaded the manuscript in doc format

-please add author contributions to your main manuscript text

Added

-please add a conflict of interest statement to your main manuscript text

added

-please use the [10 author names, et al.] format in your references (i.e. limit the author names to the first 10)

modified accordingly

Figure Issues:

-Figure 3A: The top part of the zoomed in part seems to be cut off and doesn't perfectly match the inset. Please provide the appropriate zoomed-in part.

Now we have provided appropriate zoomed-in-part. Please see the modified fig 3A.

-Figure 4C: you need boxes to indicate where the zoomed in parts are coming from. For example, C-e looks like it matches the inset of C-d, but this is not very clear in the figure. Also, what part of the figure C-f is from? (i.e. C-f is a zoomed in part of which figure?)

Now we have indicated the zoomed parts. Fig 4Ce is an independent micrograph of flagella. Please see the modified fig 4C.

-Figure 4F: you should indicate with boxes to show which part of the main image is then zoomed in

In this figure images are taken in 10x and 63x separately (not zoomed in). Now we have indicated clearly.

April 28, 2022

RE: Life Science Alliance Manuscript #LSA-2021-01329RR

Prof. Rita Tewari
University of Nottingham
School of Life Sciences
Queens Medical Centre
Nottingham NG7 2UH
United Kingdom

Dear Dr. Tewari,

Thank you for submitting your Research Article entitled "Plasmodium SAS4: basal body component of male cell which is dispensable for parasite transmission". It is a pleasure to let you know that your manuscript is now accepted for publication in Life Science Alliance. Congratulations on this interesting work.

DISTRIBUTION OF MATERIALS:

Again, congratulations on a very nice paper. I hope you found the review process to be constructive and are pleased with how the manuscript was handled editorially. We look forward to future exciting submissions from your lab.

Sincerely,
